# Incidence of anogenital warts after the introduction of the quadrivalent HPV vaccine program in Manitoba, Canada

**Christiaan H. Righolt[1,2], Karla Willows[1], Erich V. Kliewer[1,3], Salaheddin M. Mahmud[1] \***

**1** Vaccine and Drug Evaluation Centre, Department of Community Health Sciences, University of Manitoba, Winnipeg, Manitoba, Canada, **2** Children's Hospital Research Institute of Manitoba, Winnipeg, Manitoba, Canada, **3** Cancer Control Research, British Columbia Cancer Agency, Vancouver, British Columbia, Canada

\* Salah.Mahmud@gmail.com

## Abstract

### Background

The incidence of anogenital warts (AGW) decreased after the introduction of the quadrivalent human papillomavirus (qHPV) vaccine in multiple jurisdictions. We studied how comparing AGW incidence rates with different outcomes affects the interpretation of the qHPV vaccination program. To do this, we replicated multiple study designs within a single jurisdiction (Manitoba).

### Methods

We measured the incidence rates of AGW, AGW-related prescriptions, chlamydia, and gonorrhea (the latter two as sham outcomes) between 2001 and 2017 using several clinical and administrative health databases from Manitoba. We then used incidence rate ratios (IRRs) to compare, for each outcome, the rate for the 1997–1998 birth cohort (the first cohorts eligible for the publicly funded qHPV vaccination program) and the older 1995–1996 birth cohort.

### Results

AGW incidence in Manitoba dropped 72% (95% confidence interval 54–83%) among 16–18 year-old girls and 51% (14–72%) among boys after the introduction of the female-only qHPV vaccination program. Trends in AGW-related prescriptions were different from trends in AGW diagnoses as these prescriptions peaked shortly after the introduction of the publicly funded qHPV vaccine program. Chlamydia and gonorrhea incidence rates also decreased 12% (5–18%) and 16% (-1-30%), respectively, for 16–18 year-old girls.

### Conclusions

The publicly funded school-based qHPV vaccine program reduced AGW incidence in Manitoba by three-quarters in young females. AGW-related prescriptions are a poor proxy for medically attended AGW after the introduction of the publicly funded qHPV vaccination

**Data Availability Statement:** Data used in this article was derived from administrative health and social data as a secondary use. The data was provided under specific data sharing agreements

only for approved use at Manitoba Centre for Health Policy (MCHP). The original source data is not owned by the researchers or MCHP and as such cannot be provided to a public repository. The original data source and approval for use has been noted in the acknowledgments of the article. Where necessary, source data specific to this article or project may be reviewed at MCHP with the consent of the original data providers, along with the required privacy and ethical review bodies. Because this data consists of personal health data of residents of Manitoba, its disclosure is governed by The Personal Health Information Act (PHIA), which legally bans the disclosure of the source data or derived data to a public repository. Anyone wishing to access this information should contact Manitoba Health's Health Information Privacy Committee (see https://www.gov.mb.ca/health/hipc/index.html for contact info).

**Funding:** This work was supported by the Merck Investigator Studies Program (IIS #51109) with a grant to the International Centre for Infectious Diseases (ICID). The sponsor had no role in the design or conduct of the study, including but not limited to, data identification, collection, management, analysis and interpretation, or preparation, review, or approval of the results. The opinions presented in the report do not necessarily reflect those of the sponsor. SMM's work is supported, in part, by funding from the Canada Research Chair Program (#231458). There was no additional external funding received for this study.

**Competing interests:** CHR has received an unrestricted research grant from Pfizer for an unrelated study. KW does not have a financial relationship to disclose. EK has received consulting fees from Merck Canada and GlaxoSmithKline for unrelated studies. EK has received honoraria and travel expenses from Merck Canada. SMM received research funding from Assurex, GSK, Merck, Pfizer, Roche and Sanofi for unrelated studies and is/was a member of advisory boards for GSK, Merck, Sanofi and Seqirus. This does not alter our adherence to PLOS ONE policies on sharing data and materials.

**Abbreviations:** AGW, Anogenital warts; CDS, Communicable Disease Surveillance database; CI, Confidence interval; DPIN, Drug Program Information Network; HAD, Hospital Abstracts Database; HPV, Human papillomavirus; ICD, International Classification of Diseases; IRR, Incident Rate Ratio; MH, Manitoba Health; MHPR, Manitoba Population Registry; MSD, Medical Services Database; PHIN, Personal health identification number; qHPV, vaccine: Quadrivalent HPV vaccine; STI, Sexually transmitted infection.

program. Different sexual habits in adolescents are, at most, responsible for a small portion of the reduction in AGW incidence.

## Introduction

Anogenital warts (AGW) are caused by the human papillomavirus (HPV), 90% of which are caused by HPV subtypes 6 and 11 [1]. The quadrivalent HPV (qHPV) vaccine targets HPV subtypes 6 and 11, as well as 16 and 18, which cause 70% of cervical cancers [2]. In the Canadian province of Manitoba, the qHPV vaccine was introduced in September 2008 for girls as a publicly funded, three-dose, grade 6, school-based vaccination program (school grade is based on birth year in Manitoba, grade 6 children are typically 11–12 years old). The first grade 6 cohort vaccinated through this vaccination program was the 1997 birth cohort. A publicly funded, high-risk catch-up program existed between 2012 and 2014 for 19–26 year-old females who were deemed, by their health care provider, to have an increased risk of HPV infection. The routine school-based program was changed to a two-dose schedule in 2015, starting with the 2004 birth cohort. School-based male qHPV vaccination was introduced in 2016 for the 2005 birth cohort, alongside a catch-up program for the 2002–2004 birth cohorts. The program switched from using qHPV vaccine to using nonavalent HPV vaccine in June 2018.

AGW incidence rates at the population level before and after the introduction of the qHPV vaccine have been reported in numerous studies and several meta-analyses [3–5]. The literature is, however, highly heterogeneous, both in terms of the source data and AGW ascertainment. Study results cannot be directly compared when methods are different. Some studies examine the whole population of a country or region [6, 7], some studies examine clients of a specific health insurer [8], and others are limited to patients of a specific clinic [9, 10]. The AGW case definition in these studies is variable, ranging from using specific AGW diagnosis codes [7, 8], prescriptions for AGW medications [11], both diagnosis codes or prescriptions [6, 12], clinical diagnosis [9, 10], to patient's self-assessment [13]. In addition, patterns of sexual behavior may have changed around the introduction of qHPV programs, but these changes, or changes in other sexually transmitted infections (STIs), are often not examined at the population level. Physician awareness and attitudes toward AGW and its treatment may have changed as well around the introduction of population-wide qHPV vaccination programs. All these factors may have affected the reported declines in AGW, but publication of trends in incidence rates in different jurisdictions does now allow for an evaluation of these non-vaccine factors.

We aimed to study how comparing AGW incidence rates with different outcomes affects the interpretation of the qHPV vaccination program. To do this, we studied multiple outcomes in a single jurisdiction (Manitoba). We examined changes in the incidence rate while varying the way AGW is measured (diagnostic codes versus dispensed prescriptions) and by comparing these results to chlamydia and gonorrhea incidence rates (as sham outcomes not prevented by HPV vaccination).

## Methods

### Data sources

Manitoba Health (MH) is the publicly funded health insurance agency providing comprehensive health insurance, including coverage for hospital and outpatient physician services, to the province's 1.3 million residents. Coverage is universal with no eligibility distinction based on

age or income, and participation rates are very high (>99%) [14]. Insured services include hospital, physician and preventive services, and vaccinations. MH maintains several centralized, administrative electronic databases that are linkable using a unique personal health identification number (PHIN). The completeness and accuracy of MH administrative databases are well established [15, 16], and these databases have been used extensively in studies of disease surveillance and post-marketing evaluation of various vaccines and drugs [17–19].

The MH Population Registry (MHPR) tracks addresses and dates of birth, insurance coverage and death for all insured persons. Since 1971, the Hospital Abstracts database (HAD) recorded virtually all services provided by hospitals in the province, including admissions and day surgeries [15]. The data collected comprise demographic as well as diagnosis and treatment information, including primary diagnosis and service or procedure codes, coded using the International Classification of Diseases, Ninth Revision, Clinical Modification (ICD-9-CM) before April, 2004, and the ICD-10-CA (Canadian adaptation of the ICD-10) and the Canadian Classification of Health Interventions (CCI) afterwards.

The Medical Services Database (MSD), also in operation since 1971, collects similar information, based on physician fee-for-service or shadow billing, on services provided by physicians in offices, hospitals and outpatient departments across the province. The Drug Program Information Network (DPIN), in operation since 1995, records all prescription drugs dispensed to Manitoba residents, including most personal care home residents [20]. The DPIN database captures data from pharmacy claims for formulary drugs dispensed to all Manitobans even those without prescription drug insurance. Because information is submitted electronically at the "point-of-sale", the accuracy of the recorded prescription information is excellent [20]. The Communicable Disease Surveillance Database (CDS) records all cases of notifiable diseases reported by clinicians and laboratories to MH since 1992. Under the *Manitoba Public Health Act*, clinicians must report all cases of chlamydia and gonorrhea, while clinical laboratories must also report the results of any laboratory tests positive for *Chlamydia trachomatis* or *Neisseria gonorrhoeae*. The CDS database stores information on laboratory specimen type, collection date, and test results.

## Eligibility and incidence

All persons registered with MH at any point between January 1, 2001 to December 31, 2017 (the *study period*) were eligible for inclusion in this study regardless of age or gender.

We used two definitions for AGW, one based on diagnostic and procedure codes (we refer to this as AGW) and one based on prescription data (AGW-related prescriptions), to examine the potential difference in trends between both methods. We identified incident AGW episodes of care using previously described algorithms [21]. Briefly, AGW episodes of care started with a tariff code specific to condyloma in the MSD (S1 Table) or a combination of specific diagnostic and procedure codes in the HAD (S2–S4 Tables). An episode of care ended when no AGW code (S1–S5 Tables) was identified for 12 months. We identified incident AGW-related prescription episodes from the DPIN as prescriptions for podofolix (Anatomical Therapeutic Chemical Classification System code D06BB04), imiquimod (D06BB10) or sinecatechins (D06BB12). Prescription episodes started at this first dispense date and ended when no AGW-related drugs were dispensed for a 12-month period. We used the start of each episode of care as the incidence date.

We identified incident chlamydia and gonorrhea from the CDS as reported cases at least 35 days after a previous positive test for the same STI (to allow comparison with other reported rates in Manitoba [22]).

## Statistical analysis

We calculated age-standardized and age-stratified incidence rates and 95% confidence intervals (CIs) using the end-of-year MHPR as the denominator (because we use calendar year birth cohorts) and the 2006 Canadian population as the standard population.

We used a pre-post analysis to assess the effect of the 2008 introduction of the school-based qHPV vaccination program. We defined birth cohorts 1995 and 1996 as the *prior cohort* (the last cohort not enrolled in the routine school-aged vaccination program, even though these females could have been vaccinated privately or in a non-school catch-up program) and we defined birth cohorts 1997 and 1998 as the *posterior cohort*. We calculated the incidence rate ratio (IRR) between the posterior and prior cohorts for each condition at age 16–18 (the follow-up data does not extend beyond age 18 for our youngest birth cohort). We performed a sensitivity analysis with one-year (1997 versus 1996) and three-year (1997–1999 versus 1994–1996) cohorts on either side of the introduction of the vaccine program to assess the effect of birth year.

We used SAS 9.4 (SAS Institute, Cary, North Carolina) and Stata 16 (StataCorp, College Station, Texas) for all analysis. This study was approved by the University of Manitoba Research Ethics Board (REB) and by MH's Health Information Privacy Committee. The REB waived the requirement for informed consent in this retrospective study of medical records, because the data did not contain any direct identifiers.

## Results

### Trends in incidence rates

Overall age-standardized AGW incidence rates for females were stable between 2001 (119 per 100,000 person-year, 95% CI 110–128) and 2008 (118, 110–127) (Fig 1, S6 Table); the rates started a steady decline after 2008 to 81 (74–88) per 100,000 person-years in 2017. AGW incidence rates in males hovered between 120 and 150 per 100,000 from 2001 to 2017 without any discernable trend. AGW-related prescriptions increased three to five-fold between 2001 and 2009 before declining back to early 2000s levels by 2017. Chlamydia incidence doubled for females and males from 2001 to 2008 and plateaued since; gonorrhea incidence was variable, with several outbreaks during the study period.

AGW incidence in 14–18 year-old girls decreased by 85% between 2011 to 2017, from 192 (152–238) to 29 (15–51) per 100,000, (Fig 2, S7 Table). In females aged 19–23, the AGW incidence rates dropped over 50% (from 570 to 230 per 100,000) between the introduction of the vaccine in 2006 and 2017 (S8 Table). AGW incidence for males in both age groups has not shown similar strong trends, even though rates were lowest at the end of the study period. Prescriptions for AGW-related drugs generally decreased since 2009 for both genders and both age groups, although it did not follow the same pattern as AGW diagnosis rates. Incidence of both chlamydia and gonorrhea has mostly remained stable since 2008 in these age groups.

### Incidence rates for cohorts before and after the introduction of the school-based vaccination program

The incidence of AGW, but not the incidence of chlamydia and gonorrhea, is lower for the posterior cohort compared to the prior cohort (Fig 3, Table 1, S9–S11 Tables). For instance, the AGW incidence rate in 17 year-old girls dropped from 159 (87–267) per 100,000 in the 1995 birth cohort to 24 (3–88) per 100,000 in the 1998 birth cohort.

Female AGW incidence in the 16–18 year-old age group dropped by 72% (IRR is 0.28; 95% confidence interval 0.17–0.46; Table 2) for the posterior cohort compared to the prior cohort.

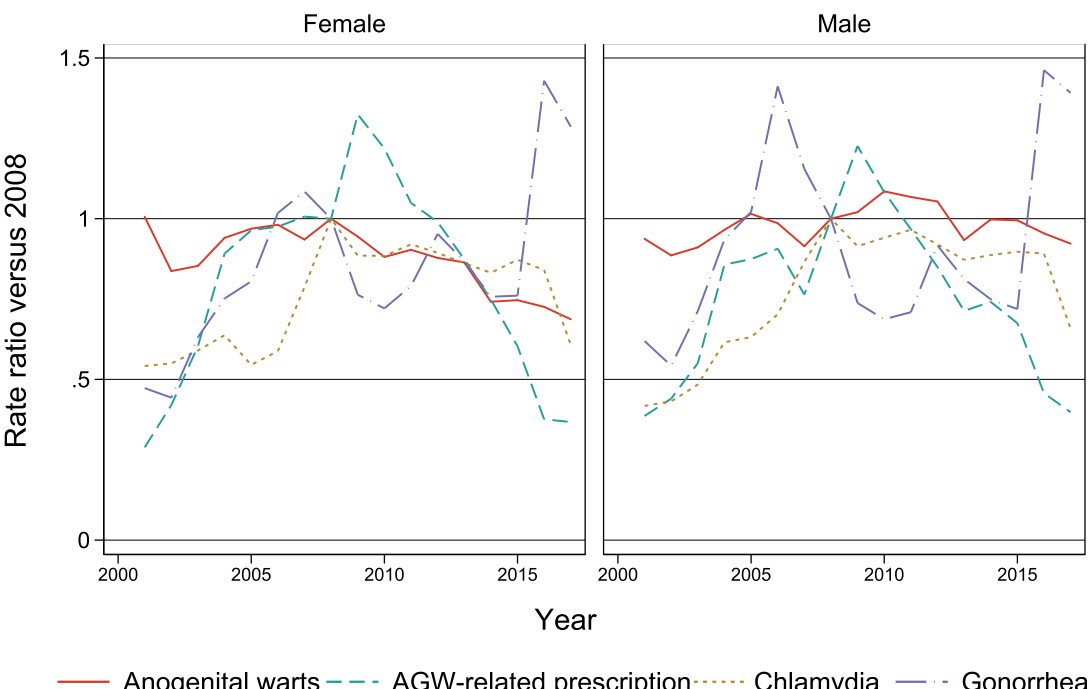

**Fig 1. Age-standardized incidence rate ratios of conditions of interest (indexed to the 2008 introduction of the school-based qHPV vaccination program) by year and gender.**

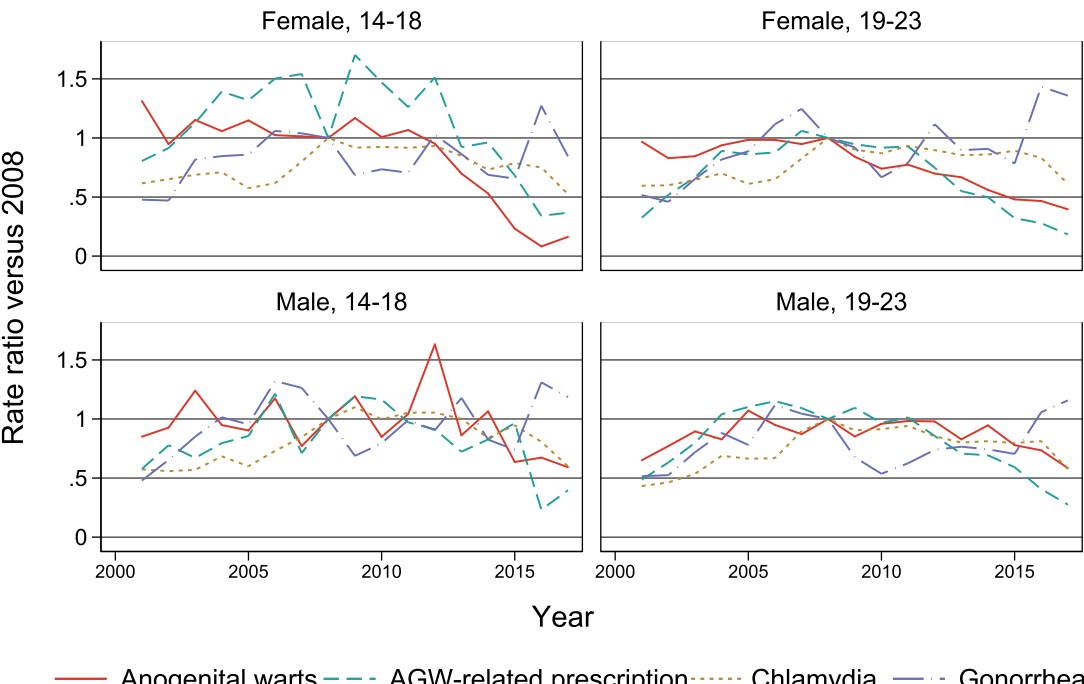

**Fig 2. Age-standardized incidence rate ratios of conditions of interest (indexed to the 2008 introduction of the school-based qHPV vaccination program) by year, age group, and gender.**

## Incidence rates among birth cohorts by age and gender

**Fig 3. Incidence rates of conditions of interest by birth cohort, age, and gender.** The 1995–1996 birth cohorts are the last cohorts not enrolled in the publicly funded school-aged vaccination program (prior cohort); the 1997–1998 birth cohorts are the first routinely enrolled cohorts (posterior cohort).

The IRR among boys was 0.49 (0.28–0.86). Males, however, were excluded from the school-based vaccination program during the study period. The change in incidence at individual ages was larger for 17 and 18 year-old girls. AGW-related prescriptions declined as well, the IRR for 16–18 year-olds was 0.39 (0.24–0.65) in girls and 0.75 (0.44–1.27) in boys. The incidence of other STIs decreased slightly as well for girls. Female chlamydia decreased by 12% (IRR is 0.88; 0.82–0.95) and gonorrhea by 16% (IRR is 0.84; 0.70–1.01).

We performed a sensitivity analysis with posterior and prior cohorts spanning either one or three years on both sides of the introduction of the qHPV vaccine program (S12, S13 Tables). The effect generally became slightly larger with increasing number of birth years per cohort. The female 16–18 year old AGW IRR is 0.34 (0.18–0.63) for one-year cohorts and 0.23 (0.15–0.34) for three-year cohorts. For chlamydia these IRRs are 0.85 (0.76–0.94) and 0.82 (0.77–0.87) respectively. This pattern does not hold for AGW-related prescriptions and gonorrhea, likely because of the relatively large fluctuations in incidence rates (Fig 1).

## Discussion

We found that AGW incidence dropped 72% (54–83%) for 16–18 year-old girls in the first cohorts eligible for the school-based qHPV vaccination program. AGW incidence in 16–18 year-old boys for these same cohorts dropped 51% (14–72%) after introduction of the female-only program. Chlamydia and gonorrhea incidence rates decreased 12% and 16%, respectively, for 16–18 year-old girls over this period. The vaccination rate in Manitoba was 52% for the first routinely eligible cohort and 72% three years later [23, 24]. The confidence bounds of the IRRs include the qHPV vaccination rates for girls as well as unvaccinated boys. Based on these results, we cannot categorically state that other effects, e.g., a herd effect, did not contribute to

**Table 1. Crude incidence rates (per 100,000 person-years; 95% confidence interval) of anogenital warts among birth cohorts by age and gender.**

| Group / birth year | 1993 | 1994 | 1995 | 1996 | 1997 | 1998 | 1999 | 2000 |
|---|---|---|---|---|---|---|---|---|
| Female 13 year-olds | 0 (0–44) | 12 (0–67) | 0 (0–44) | 0 (0–46) | 0 (0–47) | 0 (0–47) | 0 (0–47) | 13 (0–71) |
| Female 14 year-olds | 0 (0–44) | 12 (0–66) | 24 (3–86) | 12 (0–68) | 13 (0–71) | 0 (0–47) | 0 (0–47) | 0 (0–46) |
| Female 15 year-olds | 47 (13–121) | 35 (7–103) | 0 (0–43) | 36 (7–106) | 50 (14–128) | 12 (0–69) | 13 (0–70) | 25 (3–90) |
| Female 16 year-olds | 176 (99–291) | 105 (48–198) | 70 (26–152) | 107 (49–203) | 37 (8–109) | 37 (8–108) | 12 (0–69) | 0 (0–45) |
| Female 17 year-olds | 221 (133–345) | 206 (122–326) | 159 (87–267) | 176 (99–291) | 61 (20–142) | 24 (3–88) | 24 (3–88) | 12 (0–67) |
| Female 18 year-olds | 367 (251–518) | 279 (180–411) | 157 (86–263) | 184 (105–299) | 60 (19–139) | 23 (3–85) | 47 (13–120) | N/A |
| Female 19 year-olds | 432 (307–591) | 405 (285–558) | 307 (204–444) | 169 (95–279) | 46 (13–118) | 91 (39–179) | N/A | N/A |
| Female 20 year-olds | 435 (311–592) | 355 (245–499) | 258 (166–385) | 220 (134–340) | 123 (61–220) | N/A | N/A | N/A |
| Female 21 year-olds | 408 (289–560) | 309 (207–444) | 317 (214–453) | 261 (167–388) | N/A | N/A | N/A | N/A |
| Female 22 year-olds | 310 (208–445) | 379 (265–525) | 209 (128–323) | N/A | N/A | N/A | N/A | N/A |
| Female 23 year-olds | 283 (187–412) | 302 (202–433) | N/A | N/A | N/A | N/A | N/A | N/A |
| Male 13 year-olds | 0 (0–42) | 0 (0–42) | 0 (0–43) | 12 (0–65) | 12 (0–67) | 12 (0–67) | 24 (3–86) | 0 (0–44) |
| Male 14 year-olds | 11 (0–62) | 0 (0–42) | 23 (3–83) | 0 (0–43) | 0 (0–44) | 0 (0–44) | 35 (7–103) | 0 (0–44) |
| Male 15 year-olds | 22 (3–80) | 11 (0–62) | 0 (0–42) | 11 (0–63) | 12 (0–65) | 23 (3–85) | 12 (0–65) | 23 (3–85) |
| Male 16 year-olds | 33 (7–96) | 22 (3–80) | 45 (12–115) | 45 (12–115) | 23 (3–84) | 12 (0–65) | 0 (0–42) | 0 (0–43) |
| Male 17 year-olds | 65 (24–141) | 87 (38–172) | 66 (24–144) | 55 (18–129) | 68 (25–149) | 23 (3–82) | 68 (25–148) | 11 (0–63) |
| Male 18 year-olds | 86 (37–169) | 191 (113–302) | 97 (44–184) | 108 (52–198) | 44 (12–114) | 33 (7–97) | 108 (52–199) | N/A |
| Male 19 year-olds | 310 (209–443) | 114 (57–204) | 209 (128–322) | 169 (96–274) | 96 (44–182) | 74 (30–152) | N/A | N/A |
| Male 20 year-olds | 313 (213–445) | 293 (196–420) | 224 (140–339) | 244 (157–364) | 143 (78–241) | N/A | N/A | N/A |
| Male 21 year-olds | 418 (301–565) | 388 (276–530) | 431 (312–580) | 311 (211–441) | N/A | N/A | N/A | N/A |
| Male 22 year-olds | 326 (224–457) | 324 (223–456) | 189 (114–295) | N/A | N/A | N/A | N/A | N/A |
| Male 23 year-olds | 369 (261–507) | 380 (270–520) | N/A | N/A | N/A | N/A | N/A | N/A |

this decline on top of direct reductions from vaccinations. Because HPV is sexually transmitted, a potential herd effect might be further decreased with increasing number of sexual partners and increasing age, as there was no universal catch-up program in Manitoba.

Reductions of AGW incidence or prevalence over 25% have been reported in young females after the introduction of qHPV vaccine in many other Western jurisdictions [6, 12, 25–35]. These jurisdictions differ in their baseline AGW incidence, age of routine vaccination, program implementation details (e.g., the extent of catch-up programs and mode of delivery), and

**Table 2. Incidence rate ratios (95% confidence interval) of certain conditions for cohorts 2 years before and after the introduction of school-based qHPV vaccination (birth cohorts 1997–1998 vs 1995–1996) by gender.**

| Condition | 16 year-olds | 17 year-olds | 18 year-olds | 16–18 year-olds |
|---|---|---|---|---|
| Anogenital warts | | | | |
| Female | 0.42 (0.16–1.08) | 0.25 (0.11–0.58) | 0.24 (0.11–0.55) | 0.28 (0.17–0.46) |
| Male | 0.39 (0.10–1.46) | 0.75 (0.30–1.87) | 0.38 (0.16–0.90) | 0.49 (0.28–0.86) |
| AGW-related prescription | | | | |
| Female | 0.86 (0.36–2.08) | 0.44 (0.22–0.90) | 0.05 (0.01–0.41) | 0.39 (0.24–0.65) |
| Male | 1.03 (0.36–2.95) | 0.52 (0.19–1.38) | 0.81 (0.37–1.78) | 0.75 (0.44–1.27) |
| Chlamydia | | | | |
| Female | 0.79 (0.68–0.91) | 0.89 (0.78–1.00) | 0.94 (0.84–1.06) | 0.88 (0.82–0.95) |
| Male | 0.80 (0.61–1.06) | 0.98 (0.78–1.23) | 0.93 (0.77–1.13) | 0.92 (0.81–1.05) |
| Gonorrhea | | | | |
| Female | 0.73 (0.51–1.05) | 0.67 (0.49–0.91) | 1.12 (0.84–1.49) | 0.84 (0.70–1.01) |
| Male | 1.79 (0.95–3.38) | 0.66 (0.40–1.07) | 1.37 (0.93–2.01) | 1.14 (0.87–1.50) |

vaccine coverage. This heterogeneity, combined with additional heterogeneity in study designs, led to significant residual heterogeneity in a meta-analysis of AGW reduction after qHPV vaccine introduction [3], we aimed to explore how differences in study design can drive heterogeneity, especially now early reports suggest cervical cancer incidence is reduced after introduction of an HPV vaccine program [36].

The study population may affect results; we included the full Manitoba population. In a recent US study among persons insured by an integrated health care delivery system in Washington and Oregon, pre-post IRRs of 0.33 (0.28–0.39) and 0.55 (0.41–0.74) were estimated for 15–19 year-old girls and boys [37] (compared to 0.28 [0.17–0.46] and 0.49 [0.28–0.86] in Manitoba 16–18 year-olds). This seems to indicate that similar results can be obtained from different study populations, although clinic-based data can yield vastly different results (e.g., see Fig 5 in [3]).

Prescriptions for AGW-related drugs do not follow trends in medically attended AGW in Manitoba. AGW-related prescriptions peaked around the introduction of the qHPV vaccination program. As such, they are a poor proxy for AGW incidence when diagnosis data is available, because available treatment, physician preference and guidelines change over time. We were unable to ascertain clinical diagnosis of AGW (aside from the coded diagnoses), but studies using the clinical diagnosis as an outcome have come from clinic-based studies [5] and typically report larger reductions.

Sexual behavior changes over time; in the United States, for instance, millennials and the subsequent Generation Z are less sexually active in adulthood than previous generations [38, 39]. High school students had a linearly decreasing number of sexual partners between 1991 and 2015, while condom use has trended upwards during the same period [40]. We have no reason to assume these trends were different in Manitoba; both trends combined could reduce STI incidence in 16–18 year-olds. Gonorrhea rates fluctuated too much, due to outbreaks, to serve as a good comparator over a short time period. Chlamydia rates have dropped slightly for 16–18 year-olds after the introduction of the qHPV vaccination program. Nucleic acid amplified testing was introduced for urethral and cervical swabs for chlamydia and gonorrhea testing in 2006/2007, causing a peak and subsequent decline in incidence rates [41], which may partially explain a drop in chlamydia in this age group. Changing sexual habits likely play only a minor role in the reduction of AGW incidence. It is possible that the change in chlamydia and gonorrhea testing, close to the introduction of the qHPV vaccine, changed screening patterns for STIs, which may have biased our results.

## Strengths and limitations

A major strength of this study is the availability of high quality, population-based health administrative databases in Manitoba. The completeness and accuracy of the MH databases are well established [15, 16, 20]. We likely underestimated the disease's incidence rates by including only medically attended AGW [21]. Non-differential disease misclassification would not bias these relative risk ratios. It is unlikely that persons born in 1995 and 1996 have different health seeking behavior for AGW, or different physician diagnoses, than those born in 1997 and 1998. An ecological study does not infer anything about vaccine effectiveness at the individual level. A vaccine effectiveness study for Manitoba has been published separately [42]. The sensitivity analysis shows the stability of our results when adding or subtracting an additional birth year to the prior and posterior cohorts.

Because qHPV vaccination was available privately before the introduction of the publicly funded program, it is possible that some girls in the prior cohort were vaccinated. In Manitoba, around 0.5% of 10–14 year-old girls and 1.2% of 15–19 year-old girls were vaccinated each

year before the introduction of the publicly funded program [43]. A study in the neighboring province of Ontario showed that only a small fraction (about 1%) of girls born in the two years before they would have been eligible for the publicly funded qHPV program (akin to our prior cohort) received the vaccine [44]. Because the vaccination rate was 52%-72% for the prior cohort [23, 24], any potential bias in our estimates due to private vaccination is likely to be minimal.

Manitoba's qHPV vaccination program has changed, males are now vaccinated as well, and the recommended number of doses has been reduced to two for both males and females. The nonavalent HPV vaccine has recently been introduced in Canada [45], but has only been publicly funded in Manitoba since 2018. All these changes will affect AGW incidence in Manitoba, both at the population level and the individual level. The effect of these changes in the vaccine program on AGW can be evaluated once the affected cohorts age into their late teens.

## Conclusions

We found that AGW incidence in Manitoba has dropped after the introduction of the school-based qHPV vaccination program. AGW incidence dropped by three-quarters for 16–18 year-old girls in the birth cohorts directly after qHPV vaccinePVHPOV introduction and by half for boys (who were not included in the publicly funded program at that time). This reduction might be partially explained by a herd effect, although we did not find conclusive evidence for this at the population level. Because AGW-related prescriptions peak around the time the qHPV vaccination program was introduced, they are a poor proxy of incidence rates based on medically attended AGW. Different sexual habits in adolescents, as evidenced by a small drop in chlamydia incidence in the same 16–18 year-old girls, is responsible for at most a small part of the reduction. The school-based qHPV vaccination program has started to reduce AGW incidence in Manitoba significantly, most notably in females, yet the full effect will only be known once the vaccinated cohorts age into their peak incidence years.

## Supporting information

**S1 Table. Tariff codes used to identify a person with anogenital warts in the Medical Services Database.**
(PDF)

**S2 Table. Identification of a person with anogenital warts from the Hospital Abstracts database.**
(PDF)

**S3 Table. ICD-9-CM procedure codes used to assist in the identification of a person with anogenital warts.**
(PDF)

**S4 Table. ICD-10-CA procedure codes used to assist in the identification of a person with anogenital warts.**
(PDF)

**S5 Table. Tariff codes for treatment of anogenital warts used to assist in the identification of extended care for a person with anogenital warts in combination with an ICD-9 code of 078.**
(PDF)

**S6 Table. Age-standardized incidence rate per 100,000 person-years (95% confidence interval) of certain conditions by year and gender.**
(PDF)

**S7 Table. Crude incidence rate per 100,000 person-years (95% confidence interval) of certain conditions among 14–18 year-olds by year and gender.**
(PDF)

**S8 Table. Crude incidence rate per 100,000 person-years (95% confidence interval) of certain conditions among 19–23 year-olds by year and gender.**
(PDF)

**S9 Table. Crude incidence rates (per 100,000 person-years; 95% confidence interval) of AGW-related prescription among birth cohorts by age and gender.**
(PDF)

**S10 Table. Crude incidence rates (per 100,000 person-years; 95% confidence interval) of chlamydia among birth cohorts by age and gender.**
(PDF)

**S11 Table. Crude incidence rates (per 100,000 person-years; 95% confidence interval) of gonorrhea among birth cohorts by age and gender.**
(PDF)

**S12 Table. Incidence rate ratios (95% confidence interval) of certain conditions for cohorts 1 year before and after the introduction of school-based qHPV vaccination (birth cohort 1997 vs 1996) by gender.**
(PDF)

**S13 Table. Incidence rate ratios (95% confidence interval) of certain conditions for cohorts 3 years before and after the introduction of school-based qHPV vaccination (birth cohorts 1997–1999 vs 1994–1996) by gender.**
(PDF)

**S14 Table. Crude incidence rate per 100,000 person-years (95% confidence interval) of certain conditions among 15–19 year-olds by year and gender.**
(PDF)

**S15 Table. Crude incidence rate per 100,000 person-years (95% confidence interval) of certain conditions among 20–24 year-olds by year and gender.**
(PDF)

**S16 Table. Crude incidence rate per 100,000 person-years (95% confidence interval) of certain conditions among 25–29 year-olds by year and gender.**
(PDF)

**S17 Table. Crude incidence rate per 100,000 person-years (95% confidence interval) of certain conditions among 30–39 year-olds by year and gender.**
(PDF)

## Acknowledgments

The authors acknowledge the Manitoba Centre for Health Policy for use of data contained in the Population Health Research Data Repository under project # 2015–019 (HIPC # 2014/2015-46; REB # HS18467 (H2015:026); RRIC # 2015–060). The results and conclusions are

those of the authors and no official endorsement by the Manitoba Centre for Health Policy, Manitoba Health, or other data providers is intended or should be inferred. Data used in this study are from the Population Health Research Data Repository housed at the Manitoba Centre for Health Policy, University of Manitoba and were derived from data provided by Manitoba Health.

## Author Contributions

**Conceptualization:** Christiaan H. Righolt, Karla Willows, Erich V. Kliewer, Salaheddin M. Mahmud.

**Data curation:** Karla Willows.

**Formal analysis:** Christiaan H. Righolt.

**Funding acquisition:** Erich V. Kliewer, Salaheddin M. Mahmud.

**Investigation:** Salaheddin M. Mahmud.

**Methodology:** Christiaan H. Righolt, Salaheddin M. Mahmud.

**Project administration:** Christiaan H. Righolt, Salaheddin M. Mahmud.

**Resources:** Salaheddin M. Mahmud.

**Software:** Christiaan H. Righolt.

**Supervision:** Salaheddin M. Mahmud.

**Visualization:** Christiaan H. Righolt.

**Writing – original draft:** Christiaan H. Righolt, Salaheddin M. Mahmud.

**Writing – review & editing:** Karla Willows, Erich V. Kliewer, Salaheddin M. Mahmud.

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
