## [Decision Letter · Decision Letter 0]

12 Dec 2021

PONE-D-21-29641Incidence of anogenital warts after the introduction of the quadrivalent HPV vaccine program in Manitoba, CanadaPLOS ONE

Dear Dr. Mahmud,

Thank you for submitting your manuscript to PLOS ONE. After careful consideration, we feel that your manuscript is near to meeting PLOS ONE’s publication criteria as it currently stands. Please address the minor suggestion raised by Reviewer #2 and resubmit by 14 December 2021. Please include the following items when submitting your revised manuscript:A rebuttal letter that responds to each point raised by the academic editor and reviewer(s). You should upload this letter as a separate file labeled 'Response to Reviewers'.A marked-up copy of your manuscript that highlights changes made to the original version. You should upload this as a separate file labeled 'Revised Manuscript with Track Changes'.An unmarked version of your revised paper without tracked changes. You should upload this as a separate file labeled 'Manuscript'.If applicable, we recommend that you deposit your laboratory protocols in protocols.io to enhance the reproducibility of your results. Protocols.io assigns your protocol its own identifier (DOI) so that it can be cited independently in the future. For instructions see: https://journals.plos.org/plosone/s/submission-guidelines#loc-laboratory-protocols. Additionally, PLOS ONE offers an option for publishing peer-reviewed Lab Protocol articles, which describe protocols hosted on protocols.io. Read more information on sharing protocols at https://plos.org/protocols?utm_medium=editorial-email&utm_source=authorletters&utm_campaign=protocols.

We look forward to receiving your revised manuscript.

Kind regards,

R Matthew Chico, MPH, PhD

Academic Editor

PLOS ONE

2. Please ensure that you have specified (1) whether consent was informed, (2) what type you obtained (for instance, written or verbal, and if verbal, how it was documented and witnessed). If your study included minors, state whether you obtained consent from parents or guardians. If the need for consent was waived by the ethics committee and (3) If you are reporting a retrospective study of medical records or archived samples, please ensure that you have discussed whether all data were fully anonymized before you accessed them and/or whether the IRB or ethics committee waived the requirement for informed consent. If patients provided informed written consent to have data from their medical records used in research, please include this information.

“This work was supported by the Merck Investigator Studies Program with a grant to the International Centre for Infectious Diseases (ICID). The sponsor had no role in the design or conduct of the study, including but not limited to, data identification, collection, management, analysis and interpretation, or preparation, review, or approval of the results. The opinions presented in the report do not necessarily reflect those of the sponsor. SMM’s work is supported, in part, by funding from the Canada Research Chair Program.”

5. Thank you for stating the following in the Funding Section of your manuscript:

“This work was supported by the Merck Investigator Studies Program with a grant to the International Centre for Infectious Diseases (ICID). The sponsor had no role in the design or conduct of the study, including but not limited to, data identification, collection, management, analysis and interpretation, or preparation, review, or approval of the results. The opinions presented in the report do not necessarily reflect those of the sponsor. SMM’s work is supported, in part, by funding from the Canada Research Chair Program.”

“This work was supported by the Merck Investigator Studies Program with a grant to the International Centre for Infectious Diseases (ICID). The sponsor had no role in the design or conduct of the study, including but not limited to, data identification, collection, management, analysis and interpretation, or preparation, review, or approval of the results. The opinions presented in the report do not necessarily reflect those of the sponsor. SMM’s work is supported, in part, by funding from the Canada Research Chair Program.”

6. Thank you for stating the following in the Competing Interests section:

“CHR has received an unrestricted research grant from Pfizer for an unrelated study. SMM has received unrestricted research grants from Merck, GlaxoSmithKline, Sanofi Pasteur, Pfizer and Roche-Assurex for unrelated studies. SMM has received fees as an advisory board member for GlaxoSmithKline, Merck, Pfizer, Sanofi Pasteur and Seqirus. EK has received consulting fees from Merck Canada and GlaxoSmithKline for unrelated studies. EK has received honoraria and travel expenses from Merck Canada. None of the other authors has any conflicts of interest that could affect the design or analysis of this project.”

7. In your Data Availability statement, you have not specified where the minimal data set underlying the results described in your manuscript can be found. PLOS defines a study's minimal data set as the underlying data used to reach the conclusions drawn in the manuscript and any additional data required to replicate the reported study findings in their entirety. All PLOS journals require that the minimal data set be made fully available. For more information about our data policy, please see http://journals.plos.org/plosone/s/data-availability.

8. We note that you have indicated that data from this study are available upon request. PLOS only allows data to be available upon request if there are legal or ethical restrictions on sharing data publicly. For more information on unacceptable data access restrictions, please see http://journals.plos.org/plosone/s/data-availability#loc-unacceptable-data-access-restrictions.

Reviewers' comments:

Reviewer's Responses to Questions

**Comments to the Author**

1. Is the manuscript technically sound, and do the data support the conclusions?

Reviewer #1: Yes

Reviewer #2: Yes

2. Has the statistical analysis been performed appropriately and rigorously? 

Reviewer #1: Yes

Reviewer #2: Yes

3. Have the authors made all data underlying the findings in their manuscript fully available?

Reviewer #1: Yes

Reviewer #2: Yes

4. Is the manuscript presented in an intelligible fashion and written in standard English?

Reviewer #1: Yes

Reviewer #2: Yes

5. Review Comments to the Author

Reviewer #1: This is a well conducted analysis and well written report of methods and findings. Results should contribute to the established knowledge base on population level effect of HPV vaccines.

Reviewer #2: This is a very interesting, well-designed and well-written manuscript on the effect of HPV vaccination on the incidence of anogenital warts. Since now we also have data on the impact of vaccination on the incidence of cervical cancer, this data may be included in the Discussion of the manuscript.

6. PLOS authors have the option to publish the peer review history of their article (what does this mean?). If published, this will include your full peer review and any attached files.

Reviewer #1: No

Reviewer #2: **Yes: **ELECTRA NICOLAIDOU

---

## [Author Response · Author response to Decision Letter 0]

10 Jan 2022

Editorial and Reviewer comments

We thank the Editor and Reviewers for their careful review of our manuscript and for the constructive remarks and advice. We know that these reviews are done by busy scientists on pro bono basis, and we truly appreciate their valuable service. 

Replies to Editorial comments

1: Please ensure that your manuscript meets PLOS ONE's style requirements, including those for file naming.

A1: We followed the instructions in the two linked guides.

2: Please ensure that you have specified (1) whether consent was informed, (2) what type you obtained (for instance, written or verbal, and if verbal, how it was documented and witnessed). If your study included minors, state whether you obtained consent from parents or guardians. If the need for consent was waived by the ethics committee and (3) If you are reporting a retrospective study of medical records or archived samples, please ensure that you have discussed whether all data were fully anonymized before you accessed them and/or whether the IRB or ethics committee waived the requirement for informed consent. If patients provided informed written consent to have data from their medical records used in research, please include this information.

A2: We added the following to the end of the methods: “The REB waived the requirement for informed consent in this retrospective study of medical records, because the data did not contain any direct identifiers.”

3-5: We note that the grant information you provided in the ‘Funding Information’ and ‘Financial Disclosure’ sections do not match. When you resubmit, please ensure that you provide the correct grant numbers for the awards you received for your study in the ‘Funding Information’ section

Please provide an amended statement that declares *all* the funding or sources of support (whether external or internal to your organization) received during this study, as detailed online in our guide for authors at http://journals.plos.org/plosone/s/submit-now. Please also include the statement “There was no additional external funding received for this study.” in your updated Funding Statement. Please include your amended Funding Statement within your cover letter. We will change the online submission form on your behalf.

Please remove any funding-related text from the manuscript and let us know how you would like to update your Funding Statement. Please include your amended statements within your cover letter; we will change the online submission form on your behalf.

A3-5: We removed the section from the manuscript, the corrected statement, incorporating your comments above is:

This work was supported by the Merck Investigator Studies Program (IIS #51109) with a grant to the International Centre for Infectious Diseases (ICID). The sponsor had no role in the design or conduct of the study, including but not limited to, data identification, collection, management, analysis and interpretation, or preparation, review, or approval of the results. The opinions presented in the report do not necessarily reflect those of the sponsor. SMM’s work is supported, in part, by funding from the Canada Research Chair Program (#231458). There was no additional external funding received for this study.

6: Competing interest. Please confirm that this does not alter your adherence to all PLOS ONE policies on sharing data and materials, by including the following statement: ""This does not alter our adherence to PLOS ONE policies on sharing data and materials.” (as detailed online in our guide for authors http://journals.plos.org/plosone/s/competing-interests). If there are restrictions on sharing of data and/or materials, please state these. Please note that we cannot proceed with consideration of your article until this information has been declared.

A6: The corrected competing interest section (which we have removed from the manuscript should be:

CHR has received an unrestricted research grant from Pfizer for an unrelated study. KW does not have a financial relationship to disclose. EK has received consulting fees from Merck Canada and GlaxoSmithKline for unrelated studies. EK has received honoraria and travel expenses from Merck Canada. SMM received research funding from Assurex, GSK, Merck, Pfizer, Roche and Sanofi for unrelated studies and is/was a member of advisory boards for GSK, Merck, Sanofi and Seqirus. This does not alter our adherence to PLOS ONE policies on sharing data and materials.

7-8: In your Data Availability statement, you have not specified where the minimal data set underlying the results described in your manuscript can be found. PLOS defines a study's minimal data set as the underlying data used to reach the conclusions drawn in the manuscript and any additional data required to replicate the reported study findings in their entirety. All PLOS journals require that the minimal data set be made fully available. For more information about our data policy, please see http://journals.plos.org/plosone/s/data-availability.

We note that you have indicated that data from this study are available upon request. PLOS only allows data to be available upon request if there are legal or ethical restrictions on sharing data publicly. For more information on unacceptable data access restrictions, please see http://journals.plos.org/plosone/s/data-availability#loc-unacceptable-data-access-restrictions.

A7-8: The corrected Data Availability statement (which we removed from the manuscript) is:

Data used in this article was derived from administrative health and social data as a secondary use. The data was provided under specific data sharing agreements only for approved use at Manitoba Centre for Health Policy (MCHP). The original source data is not owned by the researchers or MCHP and as such cannot be provided to a public repository. The original data source and approval for use has been noted in the acknowledgments of the article. Where necessary, source data specific to this article or project may be reviewed at MCHP with the consent of the original data providers, along with the required privacy and ethical review bodies. Because this data consists of personal health data of residents of Manitoba, its disclosure is governed by The Personal Health Information Act (PHIA), which legally bans the disclosure of the source data or derived data to a public repository. Anyone wishing to access this information should contact Manitoba Health’s Health Information Privacy Committee (see https://www.gov.mb.ca/health/hipc/index.html for contact info).

Replies to Reviewer 2

Q: Since now we also have data on the impact of vaccination on the incidence of cervical cancer, this data may be included in the Discussion of the manuscript.

A: We have included this. Thank you for the suggestion.

---

## [Decision Letter · Decision Letter 1]

13 Apr 2022

Incidence of anogenital warts after the introduction of the quadrivalent HPV vaccine program in Manitoba, Canada

PONE-D-21-29641R1

Dear Dr. Mahmud,

We’re pleased to inform you that your manuscript has been judged scientifically suitable for publication and will be formally accepted for publication once it meets all outstanding technical requirements.

Kind regards,

Maria Lina Tornesello

Academic Editor

PLOS ONE

Additional Editor Comments (optional):

Reviewers' comments:

Reviewer's Responses to Questions

**Comments to the Author**

1. If the authors have adequately addressed your comments raised in a previous round of review and you feel that this manuscript is now acceptable for publication, you may indicate that here to bypass the “Comments to the Author” section, enter your conflict of interest statement in the “Confidential to Editor” section, and submit your "Accept" recommendation.

Reviewer #2: (No Response)

2. Is the manuscript technically sound, and do the data support the conclusions?

Reviewer #2: Yes

3. Has the statistical analysis been performed appropriately and rigorously? 

Reviewer #2: Yes

4. Have the authors made all data underlying the findings in their manuscript fully available?

Reviewer #2: Yes

5. Is the manuscript presented in an intelligible fashion and written in standard English?

Reviewer #2: Yes

6. Review Comments to the Author

Reviewer #2: I was not able to find the comment about the impact of vaccination on the incidence of cervical cancer, even though the authors stated that they included a comment in their revised manuscript. The authors may submit the manuscript again with the comment highlighted.

7. PLOS authors have the option to publish the peer review history of their article (what does this mean?). If published, this will include your full peer review and any attached files.

Reviewer #2: **Yes: **ELECTRA NICOLAIDOU

---

## [Editor Report · Acceptance letter]

18 Apr 2022

PONE-D-21-29641R1 

Incidence of anogenital warts after the introduction of the quadrivalent HPV vaccine program in Manitoba, Canada 

Dear Dr. Mahmud:

I'm pleased to inform you that your manuscript has been deemed suitable for publication in PLOS ONE. Congratulations! Your manuscript is now with our production department. 

Kind regards, 

on behalf of

Dr. Maria Lina Tornesello 

Academic Editor

PLOS ONE